# SPATIO-TEMPORAL GRAPH LEARNING WITH LARGE LANGUAGE MODEL

## ABSTRACT

Spatio-temporal prediction holds immense significance in urban computing as it enables decision-makers to anticipate critical phenomena such as traffic flow, crime rates, and air quality. Researchers have made remarkable progress in this field by leveraging the graph structure inherent in spatio-temporal data and harnessing the power of Graph Neural Networks (GNNs) to capture intricate relationships and dependencies across different time slots and locations. These advancements have significantly improved representation learning, leading to more accurate predictions. This study focuses on exploring the capacity of Large Language Models (LLMs) to handle the dynamic nature of spatio-temporal data in urban systems. The proposed approach, called STLLM, integrates LLMs with a cross-view mutual information maximization paradigm to capture implicit spatio-temporal dependencies and preserve spatial semantics in urban space. By harnessing the power of LLMs, the approach effectively captures intricate and implicit spatial and temporal patterns, resulting in the generation of robust and invariant LLM-based knowledge representations. In our framework, the cross-view knowledge alignment ensures effective alignment and information preservation across different views while also facilitating spatio-temporal data augmentation. The effectiveness of STLLM is evaluated through theoretical analyses, extensive experiments, and additional investigations, demonstrating its ability to align LLM-based spatio-temporal knowledge and outperform state-of-the-art baselines in various prediction tasks.

## 1 INTRODUCTION

Spatio-temporal prediction refers to the task of forecasting future events or conditions by taking into account both spatial and temporal information (Pan et al., 2019). It holds immense significance in the field of urban computing as it enables the prediction of various phenomena such as traffic flow (Zheng et al., 2020), crime rates (Wang et al., 2022a), and air quality (Liang et al., 2023). By providing these predictions, decision-makers can take proactive measures, allocate resources efficiently, and engage in effective urban planning, thereby enhancing efficiency, sustainability, and public safety in cities.

In the pursuit of achieving accurate forecasting results, researchers have made significant strides in developing innovative techniques that exploit the inherent graph structure of spatio-temporal data, enabling them to capture intricate relationships and dependencies across different time slots and locations. Notably, Graph Neural Networks (GNNs) have emerged as a powerful tool in this field, offering the ability to incorporate both spatial and temporal information into the representation learning process. Recent advancements in spatio-temporal GNNs, including graph convolutional networks (Yu et al., 2018), graph attention mechanisms (Lan et al., 2022), and graph transformers (Huo et al., 2023), have been proposed to enhance the representation learning capabilities of spatio-temporal graphs. These GNN-based techniques employ diverse embedding propagation schemes over the generated spatio-temporal graph structures to effectively capture spatial and temporal patterns.

Spatio-temporal prediction presents several challenges that require careful consideration in practical urban computing scenarios. i) **Long-Range Spatio-Temporal Dependencies**: capturing long-range dependencies is crucial as spatio-temporal phenomena often exhibit dependencies that span extensive time intervals and distant locations. This poses a challenge, as short-term local interactions may not fully reflect the underlying dynamics accurately. ii) **Data Sparsity and Noise**: data sparsity is prevalent in spatio-temporal datasets, where limited observations (*e.g.*, crimes, traffic accidents) are

available for certain locations and time intervals. Moreover, missing and noisy values in the spatio-temporal data further hinder the prediction task, often due to sensor failures or noisy connections in the constructed spatio-temporal graphs. iii) **Dynamic and Evolving Nature**: Spatio-temporal systems are inherently dynamic and subject to changes over time. Predictive models need to adapt to these dynamic characteristics by effectively distilling invariant representations with spatio-temporal semantics, enabling them to handle evolving patterns and shifts.

In this study, our focus is on exploring the application of Large Language Models (LLMs) in spatio-temporal prediction to address the aforementioned challenges. While LLMs have gained substantial attention and success in domains of NLP (Zhao et al., 2023; Ji et al., 2023) and multi-modal understanding (Yin et al., 2023; Driess et al., 2023), their exploration in forecasting spatio-temporal graph data remains relatively unexplored. This work aims to bridge this gap by harnessing the superior capabilities of LLMs in distilling intricate and implicit spatial and temporal patterns.

**Contribution**. This study presents STLLM, a new LLM-enhanced spatio-temporal learning paradigm that leverages Large Language Models to enhance the understanding of spatio-temporal data. By integrating a LLM-based spatio-temporal knowledge learner with a cross-view mutual information maximization paradigm, our approach effectively captures spatio-temporal connections and preserves point of interest information across the urban space, offering a comprehensive view of spatio-temporal features. The LLM-based knowledge can serve as robust and invariant representations, particularly in scenarios involving spatio-temporal distribution shifts. Furthermore, our spatio-temporal knowledge alignment paradigm maximizes mutual information between LLM-based knowledge representations and GNN-based structural embeddings, ensuring effective alignment and information preservation across different views. This cross-view knowledge alignment process not only facilitates effective data augmentation but also addresses inaccuracies in the raw spatio-temporal graph data by denoising noisy connections. The effectiveness of our proposed STLLM is further strengthened by theoretical analyses, demonstrating its ability to align LLM-based spatio-temporal knowledge through the maximization of mutual information. Extensive experiments are conducted to evaluate the effectiveness of STLLM in various spatio-temporal prediction tasks, comparing it with state-of-the-art baselines. Additional analyses, such as model ablation studies, robustness investigations, and efficiency evaluations, are performed to validate the efficacy of STLLM. To access the model implementation for result reproducibility, please visit the following link: https://anonymous.4open.science/r/STLLM.

## 2 RELATED WORK

**Region Representation Learning**. The representation of regions in the spatial-temporal semantic space has long been a subject of scholarly research (Wang & Li, 2017; Yao et al., 2018; Zhang et al., 2021; 2019; Fu et al., 2019; Wu et al., 2022; Zhang et al., 2023b). Specifically, Fu et al. (2019) recommend utilizing both intra- and inter-region information to enhance representations. Building upon this idea, Zhang et al. (2019) employ a collective adversarial training method. In a recent study by Zhang et al. (2021), they propose a multi-view joint learning model for region representation, which captures region correlations from various perspectives (*e.g.*, region attributes) and employs graph attention for representation learning. Conversely, Wu et al. (2022) focus on extracting traffic patterns for area representation, but their approach disregards essential POI data. To address the reliance on high-quality region graphs and the challenges associated with learning from noisy and skewed spatial-temporal data, Zhang et al. (2023b) propose an adversarial contrastive learning paradigm for robust spatial-temporal graph augmentation. Despite the effectiveness of previous methods, their reliance on structure information hinders adaptability. In this paper, we address this limitation by leveraging LLMs' global knowledge to uncover additional region global relationships, thereby enhancing the overall performance of region representation.

**Large Language Model**. LLMs extensively trained on large corpus, have demonstrated exceptional performance in NLP tasks (Ji et al., 2023; Wang et al., 2022b). Primarily based on the Transformer architecture (Vaswani et al., 2017), these models can be classified into three categories: encoder-only, encoder-decoder, and decoder-only (Pan et al., 2023). (i) Encoder-only LLMs utilize only the encoder for word associations and encoding sentences, like BERT (Devlin et al., 2018; Liu et al., 2019; Lan et al., 2019), requiring an additional prediction head for downstream tasks. These models excel in tasks demanding full sentence understanding (Zhang et al., 2022). (ii) Encoder-decoder LLMs employ both encoder and decoder modules for input encoding (Raffel et al., 2020; Zeng et al., 2023)

Figure 1: Architecture of our proposed STLLM framework.

and output generation, offering more flexible training techniques (Zoph et al., 2022; Xue et al., 2020). (iii) Decoder-only LLMs rely solely on the decoder module for output generation, with training centered around predicting subsequent words. Models like Chat-GPT (Ouyang et al., 2022) and GPT-4 can often complete tasks with minimal sample or instruction input. However, their closed-source nature limits further research. Recently, open-source models such as Alpaca and Vicuna have shown comparable performance (Touvron et al., 2023). This study applies decoder-only LLM (GPT-3.5) to enhance the quality of the spatio-temporal graph with effective augmentation.

## 3 METHODOLOGY

This section elaborate on the technical details of the proposed STLLM. The model architecture is depicted in Figure 1. We begin with an introduction to the spatio-temporal prediction task as follows.

### 3.1 PRELIMINARIES

The urban space is partitioned into $I$ spatial regions, indexed by $i$, and $J$ time slots, indexed by $j$. Each region is denoted as $r_i$, and each time slot is denoted as $t_j$. To facilitate comprehensive spatio-temporal (ST) representation learning, we construct a spatio-temporal graph by incorporating urban contextual information from diverse data sources. Specifically, we utilize the following data:

**i) Human Mobility Trajectories** $\mathcal{M}$. This data comprises real human mobility trajectories, where each trajectory is represented as $(r_s, r_d, t_s, t_d, v)$. Here, $r_s$ and $r_d$ denote the source and target regions, respectively, while $t_s$ and $t_d$ refer to the corresponding timestamps. $v$ denotes the mobility volume of this trajectory. These trajectories capture the temporal region-wise connections in terms of human mobility, making them essential for various urban prediction tasks. **ii) Region-wise Distance Information** $\mathcal{D}$. This data includes a weighted adjacency matrix that records neighborhood information based on region-wise distances. It encompasses all pairs of regions $(r_s, r_d, d)$ with a physical distance of less than 2.5km, and $d$ denotes the distance in kilometers. This data provides valuable spatial contextual information about urban regions, facilitating spatial analysis and modeling.

**Spatio-Temporal Graph**. Leveraging the foregoing data, we construct the spatio-temporal graph $\mathcal{G} = (\mathcal{V}, \mathcal{E})$. The node set $\mathcal{V}$ consists of $J$ time-slot-specific copies for each of the $I$ regions, resulting in a total of $|\mathcal{V}| = I \times J$ nodes. The weighted edge set $\mathcal{E}$ incorporates the two heterogeneous data sources, along with residual connections across adjacent time slots. Using $r_i t_j$ to represent the node for the $r_i$-th region in the $t_j$-th time slot, we define the edge set as follows:

$$\mathcal{E} = \{(r_s t_s, r_d t_d, v) : \mathcal{M}\} \cup \{(r_s t_j, r_d t_j, d) : \mathcal{D}, \forall t_j\} \cup \{(r_i t_j, r_i t_{j+1}, 1) : t_j < J, \forall r_i\} \quad (1)$$

**Problem Statement**: Given the spatio-temporal graph $\mathcal{G}$ constructed from the heterogeneous data, our objective in spatio-temporal representation learning is to generate an embedding matrix $\mathbf{E} \in \mathbb{R}^{|\mathcal{V}| \times d}$. Each row vector $\mathbf{e}_{ij} \in \mathbb{R}^d$ in $\mathbf{E}$ refers to the representation vector for the $r_i$-th region in the $t_j$-th time slot. This learned embedding facilitates accurate predictions in various types of downstream tasks in urban scenarios, such as traffic prediction, crime prediction, and house price prediction.

### 3.2 DUAL-VIEW SPATIO-TEMPORAL MODELING

#### 3.2.1 SPATIO-TEMPORAL GRAPH NEURAL NETWORK

Our STLLM framework incorporates two modeling views to capture spatio-temporal patterns. The first view focuses on extracting the high-order connectivity of the spatio-temporal graph through iterative graph neural propagation. It begins by generating an initial embedding for each node $v_i \in \mathcal{V}$

by projecting its node-specific point of interest (POI) tags into a $d$-dimensional latent representation $\mathbf{h}_i^0 \in \mathbf{H}^0$ using a transformer-based neural language model (Vaswani et al., 2017). Subsequently, leveraging the graph structures in the aforementioned ST graph $\mathcal{G}$, STLLM performs multiple GNN iterations to capture and refine the spatio-temporal dependencies as follows:

$$\mathbf{H} = \sum_{l=0}^{L} \mathbf{H}^l, \quad \mathbf{H}^l = \sigma(\mathbf{D}^{-1/2}\mathbf{A}\mathbf{D}^{-1/2}\mathbf{H}^{l-1}\mathbf{W}^{l-1}), \quad \mathbf{D}_{i,j} = |\mathcal{N}_i| \text{ if } i = j \text{ otherwise } 0 \quad (2)$$

where $\mathbf{H} = \{\mathbf{h}_i\} \in \mathbb{R}^{|\mathcal{V}| \times d}$ denotes the final embedding matrix given by the graph modeling view of our STLLM. $\mathbf{A} \in \mathbb{R}^{|\mathcal{V}| \times |\mathcal{V}|}$ denotes the adjacent matrix of the ST graph $\mathcal{G}$. This spatio-temporal GNN aggregates the embeddings of different propagation iterations. Each iteration is done by the graph convolution operator with the learnable linear projection $\mathbf{W}^{l-1} \in \mathbb{R}^{d \times d}$ and the ReLU-based non-linear projection $\sigma(\cdot)$. $\mathbf{D}$ denotes the symmetric degree matrix of the adjacent matrix $\mathbf{A}$.

### 3.2.2 LLM-BASED SPATIO-TEMPORAL KNOWLEDGE LEARNING

Drawing inspiration from the LLM's ability to understand real-world knowledge, we propose leveraging a well-trained LLM to generate semantic node representations. Specifically, STLLM first generates a text-based description $\mathcal{P}_i = (r_i, \mathcal{Q}_i, \mathcal{S}_i, \mathcal{T}_i)$ for each region $r_i$. This description is constructed by concatenating the region ID $r_i$, its point of interest (POI) information $\mathcal{Q}_i$, the spatial context $\mathcal{S}_i$ derived from distance data $\mathcal{D}$, and the temporal context $\mathcal{T}_i$ derived from mobility data $\mathcal{M}$.

Using these text descriptions, STLLM proceeds with two steps to acquire LLM-based knowledge representations. Firstly, we prompt the pretrained LLM to generate a summary for each node. This involves inputting the descriptions of the target node and its surrounding nodes together into the LLM to facilitate comprehension of the spatio-temporal context. Secondly, STLLM obtains latent representation vectors $\mathbf{F} = \mathbf{f}_i \in \mathbb{R}^{I \times d}$ for the summary text of the regions. Leveraging the LLM's profound understanding of general-purpose knowledge, the generated embeddings in $\mathbf{F}$ effectively preserve the POI information within each region and capture the spatio-temporal connections to its neighborhood. This approach, in contrast to the GNN-based modeling view that focuses on local structure extraction, represents the spatio-temporal features from a global perspective by distilling general knowledge from the LLM. Appendix A.5 provides examples of descriptions and summaries.

### 3.3 CROSS-VIEW MUTUAL INFORMATION MAXIMIZATION

With the GNN-based ST dependencies modeling and the LLM-based global knowledge mining, we aim at maximizing their mutual information to minimizes their respective noise and irrelevant information. To do so, STLLM is tuned utilizing the following cross-view mutual information maximization objective, with $I(\cdot)$ denoting the mutual information function:

$$\mathbf{H} = \arg\max_{\mathbf{H}} I(\mathbf{h}, \mathbf{f}), \quad \text{where } I(\mathbf{h}, \mathbf{f}) = \sum_{\mathbf{h}_i, \mathbf{f}_i} p(\mathbf{h}_i, \mathbf{f}_i) \log \frac{p(\mathbf{h}_i | \mathbf{f}_i)}{p(\mathbf{h}_i)} \quad (3)$$

To make this loss function tractable, we follow Oord et al. (2018) to utilize the infoNCE loss function for optimization, which is proved to be the lowerbound of the mutual information function. Specifically, the infoNCE loss between embeddings from the two views is defined as follows:

$$\mathcal{L}_{NCE} = -\mathbb{E}_{\mathbf{h}}[\log \frac{g(\mathbf{h}, \mathbf{f})}{\sum_{\mathbf{h}_{i'} \in \mathbf{H}} g(\mathbf{h}_{i'}, \mathbf{f})}] = -\sum_{\mathbf{h}_i \in \mathbf{H}} \log \frac{g(\mathbf{h}_i, \mathbf{f}_i)}{\sum_{\mathbf{h}_{i'} \in \mathbf{H}} g(\mathbf{h}_{i'}, \mathbf{f}_i)} \quad (4)$$

where $g(\mathbf{h}, \mathbf{f}) \propto p(\mathbf{h}|\mathbf{f})/p(\mathbf{h})$ denotes some measurement for the probability ratio. We show that with this function $g(\cdot)$, the tractable infoNCE loss $\mathcal{L}_{NCE}$ is a lowerbound of the mutual information between embeddings of the GNN view $\mathbf{h}$ and embeddings of the LLM view $\mathbf{f}$, as follows:

$$\mathbb{E}_{\mathbf{h}}[\log \frac{g(\mathbf{h}, \mathbf{f})}{\sum_{\mathbf{h}_{i'} \in \mathbf{H}} g(\mathbf{h}_{i'}, \mathbf{f})}] \propto \mathbb{E}_{\mathbf{h}}[\log \frac{\frac{p(\mathbf{h}|\mathbf{f})}{p(\mathbf{h})}}{\sum_{\mathbf{h}_{i'} \in \mathbf{H}} \frac{p(\mathbf{h}_{i'}|\mathbf{f})}{p(\mathbf{h}_{i'})}}] \approx -\mathbb{E}_{\mathbf{h}} \log[1 + \frac{p(\mathbf{h})}{p(\mathbf{h}|\mathbf{f})}(N - 1)]$$

$$\leq -\mathbb{E}_{\mathbf{h}} \log[\frac{p(\mathbf{h})}{p(\mathbf{h}|\mathbf{f})} N] = I(\mathbf{h}, \mathbf{f}) - \log(N) \quad (5)$$

## 3.4 Spatio-Temporal Knowledge Alignment

Based on the foregoing discussions, we apply the infoNCE loss to the alignment between the two modeling views, by instantiating $g(\mathbf{h}, \mathbf{f}) = \exp \cos(\mathbf{h}, \mathbf{f})$. Specifically, STLLM aligns the representation for the mobility data and for the distance data, respectively. And we further enrich the training objective of our ST representation learning framework with two consine-based loss terms: the alignment between the overall embeddings from the two views, and the alignment between the shallow and the deep GNN embeddings. In total, we have the following four training objectives:

$$\mathcal{L}_{\mathcal{M}} = \sum_{v_i \in \mathcal{V}} \mathcal{L}_{NCE}(\mathbf{h}_i^{\mathcal{M}}, \mathbf{f}_i^{\mathcal{M}}), \quad \mathcal{L}_{\mathcal{D}} = \sum_{v_i \in \mathcal{V}} \mathcal{L}_{NCE}(\mathbf{h}_i^{\mathcal{D}}, \mathbf{f}_i^{\mathcal{D}})$$

$$\mathcal{L}_{\mathcal{M},\mathcal{D}} = \sum_{v_i \in \mathcal{V}} \cos(\mathbf{h}_i, \mathbf{f}_i), \quad \mathcal{L}_{\mathcal{G}} = \sum_{v_i \in \mathcal{V}} \cos(\mathbf{h}_i, \mathbf{h}_i^0) \quad (6)$$

Combining the above four loss terms, we obtain the final loss for our STLLM: $\mathcal{L} = \gamma_1 \mathcal{L}_{\mathcal{M},\mathcal{D}} + \gamma_2 \mathcal{L}_{\mathcal{G}} + \gamma_3 \mathcal{L}_{\mathcal{M}} + \gamma_4 \mathcal{L}_{\mathcal{D}}$, where $\gamma_1, \gamma_2, \gamma_3, \gamma_4 \in \mathbb{R}$ denote the hyperparameters for the loss weights.

**Model Complexity**: Our method, STLLM, incorporates a spatio-temporal graph neural network that involves graph information propagation in the encoder. The time complexity of our method is determined by the graph operations and is given by $\mathcal{O}(|\mathcal{E}| \times L \times d)$, where $|\mathcal{E}|$ denotes the number of edges in the graph, $L$ represents the number of graph layers, and $d$ is the dimensionality of the embeddings. It is important to note that the LLM-based generation is performed only once and is not counted in the time complexity. The resulting time complexity is comparable to that of other state-of-the-art methods, ensuring efficient computation and maintaining competitive performance.

## 4 Evaluation

Our experiments investigate the following research questions: **RQ1**: How does STLLM compare to state-of-the-art baselines in different spatio-temporal learning applications, such as traffic prediction and crime prediction? **RQ2**: How do different data sources and model components impact the effectiveness of region representation learning for downstream tasks? **RQ3**: To what extent is STLLM successful in representation learning for predicting traffic flows and crimes, considering varying degrees of data sparsity? **RQ4**: What impact do various hyperparameter settings have on the region representation performance of STLLM for traffic flow and crime prediction? **RQ5**: How efficient is STLLM compared to other baseline methods? **RQ6**: How effective is STLLM compared to other techniques of region representation, such as MV-PN, MVURE, MGFN, and GraphST?

### 4.1 Experimental Setup

#### 4.1.1 Datasets and Evaluation Metrics

We assess the performance of our spatio-temporal representation learning framework, STLLM, on three distinct prediction tasks: crime prediction, traffic flow forecasting, and property price prediction. These tasks are evaluated using real-world datasets obtained from Chicago and NYC. Following previous studies (Xia et al., 2021), we consider multiple crime types such as Theft, Battery, Assault, and Damage for Chicago, and Burglary, Larceny, Robbery, and Assault for NYC. Table 3 in appendix provides detailed statistics of the datasets used. We utilize three evaluation metrics: Mean Absolute Error (MAE), Mean Absolute Percentage Error (MAPE), and Root Mean Squared Error (RMSE).

#### 4.1.2 Implementation Details and Hyperparameters

For our STLLM, we fix the dimensionality $d$ to 96 in accordance with earlier region representation studies (Wu et al., 2022; Zhang et al., 2023b). Our STLLM performs at its optimum when the GCN depth is set to 2, the weight decay is set to 0.0005, and the learning rate is set to 0.001, according to the hyperparameter trials. Following previous studies (Zhang et al., 2023a;b), we employ different downstream models for different prediction tasks. For crime prediction, we utilize ST-SHN (Xia et al., 2022). For traffic flow forecasting, we employ ST-GCN (Yu et al., 2018). For property price prediction, we utilize a simple Lasso regression (Ranstam & Cook, 2018). ST-SHN is configured with 0.001 learning rate, 0.96 learning rate decay, 2 spatial aggregation layers. ST-GCN is configured

with 12 input time intervals with a 5-minute interval length. And the traffic prediction task aims at predicting the future 15 minutes. The majority of baselines are implemented with their released code.

### 4.1.3 BASELINES FOR COMPARISON

We compare our method STLLM with various state-of-art baselines to evaluate performance in terms of MAE, MAPE and RMSE. Because of page limits, detailed descriptions for each method are showed in Appendix A.3. The baselines include the following categories. Graph Representation Methods: We compare our method STLLM with several graph representation methods, including Node2vec (Grover & Leskovec, 2016), GCN (Kipf & Welling, 2017), GraphSage (Hamilton et al., 2017), GAE (Kipf & Welling, 2016) and GAT (Veličković et al., 2018). Graph Contrastive Learning Methods: We also conduct expriments via two recent graph contrastive learning methods, GraphCL (You et al., 2020) and RGCL (Li et al., 2022). Spatio-Temporal Region Representation: We compare our method STLLM with state-of-art region representation learning methods including HDGE (Wang & Li, 2017), ZE-Mob (Yao et al., 2018), MV-PN (Fu et al., 2019), CGAL (Zhang et al., 2019), MVURE (Zhang et al., 2021), MGFN (Wu et al., 2022) and GraphST (Zhang et al., 2023b).

### 4.2 MODEL EFFECTIVENESS (RQ1)

We compare the performance of STLLM with state-of-the-art baselines across various downstream tasks. The results are presented in Table 1, based on which we draw the following discussions:

**Consistent Performance Superiority across Tasks**. Our STLLM framework surpasses all baselines across distinct research lines, demonstrating excellent performance due to the effective distillation of global spatial-temporal knowledge from LLMs. This knowledge is incorporated into the local ST graph modeling process via contrastive learning, conferring several benefits. It allows precise comprehension and use of textual ST features, improving prediction accuracy. It also increases robustness against structural noises in the ST graph. Significant enhancements are observed across all three tasks: traffic prediction, crime prediction, and house price prediction, attesting to the general applicability of our LLM-based ST graph mining techniques.

**Advantages of Graph Contrastive Learning**. Among the baseline methods, those utilizing graph contrastive learning (GCL) techniques, such as GraphCL and GraphST, exhibit notable advantages in terms of accuracy compared to other baselines. This validates the effectiveness of GCL in addressing data deficiency issues, such as noise and skewness, thereby improving the accuracy of spatio-temporal prediction. Drawing inspiration from this advantage, our STLLM incorporates the GCL method to maximize the mutual information between LLM-based ST knowledge mining and local ST graph modeling. By contrasting the two views, our model effectively enhances the representation quality.

**Advantages of Region Representation Learning**: When compared to end-to-end spatio-temporal prediction methods, pretrained region representation learning approaches (e.g., MGFN) demonstrate clear advantages. These advantages can be attributed to the superiority of pretrained embeddings over randomly-initialized embeddings. Pretrained embeddings are enriched with abundant spatial and temporal patterns, providing a more refined and informative initialization compared to end-to-end models. The limited optimization steps in end-to-end models make it challenging to acquire such sophisticated embeddings during training, leading to the sub-optimal performance.

### 4.3 ABLATION STUDY (RQ2)

In this section, we conduct ablation study to investigate the influence of different components of our STLLM. Specifically, we study the following variants: **-CL**. This version replaces the contrastive learning with the cosine similarity in the cross-view mutual information maximization. **-S**. This variant removes the spatial information $\mathcal{S}_i$ from in the input description $\mathcal{P}_i$ for the LLM, to study the influence of textual spatial features. **-T**. Similar to the last one, this variant eliminates the temporal information $\mathcal{T}_i$ from $\mathcal{P}_i$. **-S&T** removes both the spatial descriptions and the temporal descriptions for the LLM. Based on the results depicted in Figure 2, we draw the following conclusions.

**Effectiveness of Contrastive Learning**. The evaluation results reveal that in many scenarios, replacing InfoNCE-based contrastive learning with maximizing cosine similarity leads to a significant decline in performance. This observation highlights the effectiveness of our contrastive learning (CL)-based design, which derives its advantage from its close theoretical relationship with mutual

Table 1: Overall performance comparison in urban three tasks.

| Model | Crime Prediction | | | | Traffic Prediction | | | | | | House Price Prediction | | | |
|---|---|---|---|---|---|---|---|---|---|---|---|---|---|---|
| | CHI-Crime | | NYC-Crime | | CHI-Taxi | | NYC-Bike | | NYC-Taxi | | CHI-House | | NYC-House | |
| | MAE | MAPE | MAE | MAPE | MAE | RMSE | MAE | RMSE | MAE | RMSE | MAE | MAPE | MAE | MAPE |
| ST-SHN | 2.0373 | 0.9996 | 3.6740 | 0.9997 | – | – | – | – | – | – | – | – | – | – |
| ST-GCN | – | – | – | – | 0.0898 | 0.5285 | 0.0358 | 0.0480 | 0.0385 | 0.0495 | – | – | – | – |
| Node2vec | 1.6972 | 0.9013 | 3.1494 | 0.8027 | 0.0874 | 0.5196 | 0.0345 | 0.0467 | 0.0360 | 0.0490 | 12148.3202 | 32.8987 | 4789.6429 | 13.0938 |
| GCN | 1.6653 | 0.8915 | 3.0852 | 0.7248 | 0.0849 | 0.4953 | 0.0327 | 0.0428 | 0.0351 | 0.0472 | 12054.8121 | 31.3830 | 4764.6871 | 12.7329 |
| GAT | 1.6421 | 0.8716 | 3.0935 | 0.7473 | 0.0852 | 0.5036 | 0.0342 | 0.0439 | 0.0347 | 0.0469 | 11983.4383 | 29.7822 | 4753.8454 | 11.5405 |
| GraphSage | 1.6055 | 0.8697 | 3.0991 | 0.7252 | 0.0842 | 0.4976 | 0.0331 | 0.0437 | 0.0366 | 0.0492 | 11894.3823 | 28.3732 | 4739.4906 | 11.8970 |
| GAE | 1.5972 | 0.8576 | 3.0757 | 0.7357 | 0.0837 | 0.4893 | 0.0324 | 0.0420 | 0.0336 | 0.0457 | 11847.4378 | 28.4743 | 4716.4906 | 12.3987 |
| GraphCL | 1.1957 | 0.5736 | 2.5743 | 0.5875 | 0.0776 | 0.4335 | 0.0278 | 0.0366 | 0.0275 | 0.0367 | 10782.3711 | 24.1711 | 4678.8939 | 12.3803 |
| RGCL | 1.1089 | 0.5421 | 2.5672 | 0.5781 | 0.0754 | 0.4282 | 0.0254 | 0.0361 | 0.0270 | 0.0361 | 10262.9604 | 23.4849 | 4602.2038 | 11.2930 |
| HDGE | 1.4481 | 0.8133 | 2.8137 | 0.6728 | 0.0792 | 0.4697 | 0.0296 | 0.0398 | 0.0284 | 0.0355 | 10738.4378 | 25.8290 | 4679.8239 | 11.9808 |
| ZE-Mob | 1.4965 | 0.8201 | 2.8090 | 0.6616 | 0.0806 | 0.4748 | 0.0312 | 0.0417 | 0.0326 | 0.0419 | 107859.0494 | 26.0239 | 4654.1917 | 10.9033 |
| MV-PN | 1.3462 | 0.8019 | 2.7431 | 0.6606 | 0.0787 | 0.4517 | 0.0299 | 0.0402 | 0.0315 | 0.0474 | 10362.2637 | 24.2733 | 4646.6755 | 11.2351 |
| CGAL | 1.3315 | 0.7955 | 2.7362 | 0.6665 | 0.0773 | 0.4492 | 0.0290 | 0.0397 | 0.0286 | 0.0414 | 10983.9058 | 26.3947 | 4658.8221 | 11.0988 |
| MVURE | 1.2772 | 0.7285 | 2.6258 | 0.6035 | 0.0763 | 0.4483 | 0.0289 | 0.0377 | 0.0261 | 0.0349 | 10573.3678 | 25.3784 | 4638.9010 | 11.1276 |
| MGFN | 1.2689 | 0.6943 | 2.5829 | 0.5997 | 0.0759 | 0.4335 | 0.0268 | 0.0367 | 0.0256 | 0.0339 | 10463.7834 | 25.4762 | 4623.4184 | 10.8816 |
| GraphST | 1.0539 | 0.4943 | 2.5454 | 0.5675 | 0.0748 | 0.4226 | 0.0260 | 0.0347 | 0.0263 | 0.0312 | 90492.2723 | 22.1827 | 4578.9023 | 9.6732 |
| *STLLM* | 1.0481 | 0.4808 | 2.5243 | 0.5318 | 0.0739 | 0.4137 | 0.0253 | 0.0317 | 0.0236 | 0.0292 | 89271.5857 | 20.2836 | 4552.8932 | 8.9766 |

Figure 2: Ablation study of STLLM on crime prediction

information maximization. Compared to cosine similarity, InfoNCE incorporates negative relation learning to promote a beneficial uniform embedding distribution, enhancing its prediction accuracy.

**Benefits Brought by $\mathcal{S}_i$ and $\mathcal{T}_i$.** The results show that the removal of either the spatial information $\mathcal{S}_i$ or the temporal information $\mathcal{T}_i$ causes a notable performance decline. This finding not only validates the effectiveness of our LLM-based textual feature extraction but also confirms the positive impact of incorporating distance information and mobility trajectories in our global knowledge mining.

**Comparing -S&T with -S and -T**. In some cases (e.g., NYC-Larceny, CHI-Assault), removing both $\mathcal{S}_i$ and $\mathcal{T}_i$ yields even better performance than removing only one of them. This observation can be attributed to the bias effect that arises when utilizing only a single data source for knowledge mining. In such cases, the LLM may be misled by the limited information, resulting in sub-optimal representations. By constructing comprehensive ST descriptions, our STLLM avoids such circumstances.

## 4.4 PERFORMANCE OVER SPARSE DATA (RQ3)

In this study, we examine the robustness of our STLLM framework when applied to sparse spatio-temporal data for crime prediction. Specifically, we divide the regions of NYC and Chicago into two sets based on their density degrees. The density degree is determined as the ratio of time slots with non-zero crime cases to the total number of time slots for each region. The two sets of regions correspond to density ratio ranges of (0.0, 0.25] and (0.25, 0.5], respectively. To assess the performance of our STLLM method and compare it to six representative baselines, we present the results in Figure 3. From the results, we have two major observations as discussed below.

**Firstly**, it is observed that all methods achieve lower MAE values on the subset with higher data density. This finding confirms that sparse data has a detrimental effect on both representation learning and accurate spatio-temporal prediction. The limited amount of data available in sparse regions results in reduced supervision signals, which in turn leads to suboptimal model training. **Secondly**, our STLLM consistently maintains its superior performance across different levels of data sparsity. This can be attributed to the enrichment of representations through the distillation of global knowledge

from the LLM. Additionally, the contrastive knowledge alignment employed by our method enhances the supervision signals, thereby facilitating effective model training.

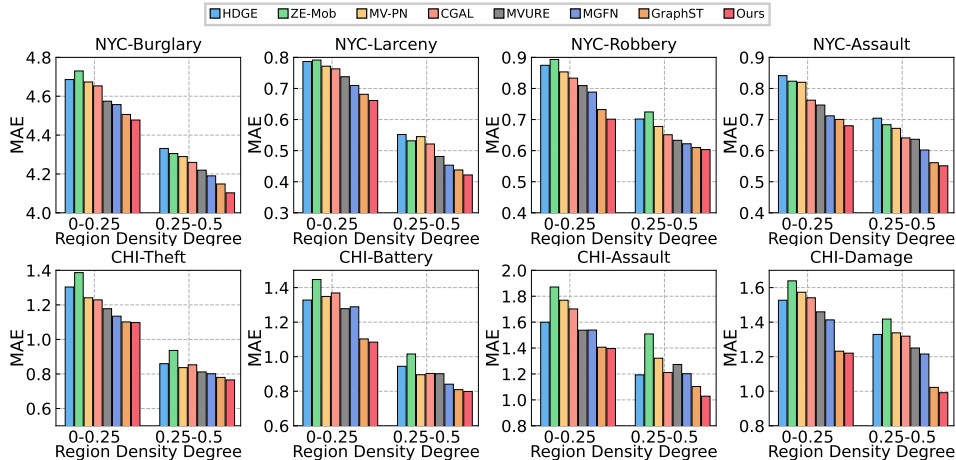

Figure 3: Results on NYC and CHI crime for four crime types *w.r.t* different data density degrees.

## 4.5 HYPERPARAMETER STUDY (RQ4)

In this section, we conducted a parameter study to evaluate the impact of important parameters on the performance of our model, STLLM. The results are presented in Figure 4. Specifically, we vary the number of GNN layers $l$ within the range $\{2, 3, 4, 5\}$ and the temperature coefficient $\tau$ in the InfoNCE function within the range $\{0.3, 0.4, 0.5, 0.6\}$. We summarize our observations regarding the two parameters and their effect on the downstream performance of traffic prediction as follows:

**Firstly**, we investigated the impact of the number of GCN layers ($l$) on the model's performance. We found that our STLLM achieves the best performance with $l = 2$. As the number of GCN layers increases beyond this point, we observed diminishing returns in terms of model representation ability for downstream tasks. This suggests that additional GCN layers may lead to an over-smoothing effect, which hampers the model's performance. **Secondly**, we examined the effect of the temperature parameter ($\tau$) on the model's representation ability. We observed that our STLLM achieves the highest representation ability with $\tau = 0.4$. Deviating from this optimal value, either by increasing or decreasing $\tau$, did not result in further improvements in the model's representation ability.

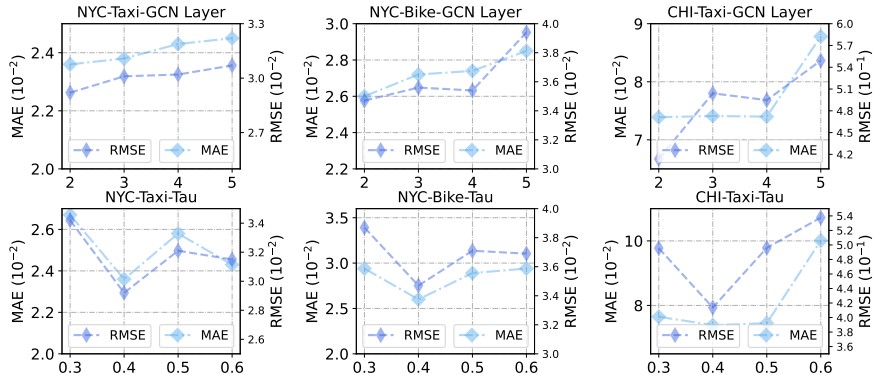

Figure 4: Hyperparameter study on traffic prediction and crime prediction *w.r.t* MAE and RMSE

## 4.6 EFFICIENCY STUDY (RQ5)

**Experimental Settings**: We assess the model efficiency of STLLM by comparing it with several region representation methods in terms of training time. The results, including MAE and MAPE values, can be found in Table 2. All methods are implemented using the same software environment (Python 3.7, TensorFlow 1.15.3 for traffic prediction, and PyTorch 1.7.0 for other tasks) and hardware

environment (a 10-core Intel Core i9-9820 CPU@3.30Hz, 64GB RAM, and four NVIDIA GeForce RTX 3090 GPUs). **Observations and Analysis**: It is worth noting that, our STLLM model achieves the best performance while maintaining efficiency comparable to other region representation methods. This finding validates that our STLLM framework is capable of handling large-scale spatio-temporal data. The scalability of STLLM can be attributed to the efficiency of the InfoNCE-based contrastive learning, which plays an important role in the process of knowledge alignment.

Table 2: The training time and performance of our STLLM and the state-of-the-art spatio-temporal region representation methods for the crime prediction task in the NYC and Chicago datasets.

| Models | HDGE | ZE-Mob | MV-PN | CGAL | MVURE | MGFN | GraphST | STLLM | HDGE | ZE-Mob | MV-PN | CGAL | MVURE | MGFN | GraphST | STLLM |
|---|---|---|---|---|---|---|---|---|---|---|---|---|---|---|---|---|
| Training | 298.5 | 79.4 | 28.7 | 4077.6 | 220.1 | 843.5 | 259.3 | 122.4 | 347.8 | 102.4 | 38.4 | 5273.8 | 280.2 | 108.9 | 334.7 | 158.1 |
| MAE | 2.8137 | 2.8090 | 2.7431 | 2.7362 | 2.6258 | 2.5829 | 2.5454 | 2.5243 | 1.4481 | 1.4965 | 1.3462 | 1.3315 | 1.2772 | 1.2689 | 1.0539 | 1.0481 |
| MAPE | 0.6728 | 0.6616 | 0.6606 | 0.6665 | 0.6035 | 0.5997 | 0.5675 | 0.5318 | 0.8133 | 0.8201 | 0.8019 | 0.7955 | 0.7285 | 0.6943 | 0.4943 | 0.4808 |

### 4.7 CASE STUDY (RQ6)

We conduct a case study to demonstrate the ability of our STLLM to learn global region dependency in terms of geographical semantics, as depicted in Figure 5. Specifically, we select two pairs of regions for analysis: nearby region pairs (such as region 170 and region 164) and faraway region pairs (such as region 144 and region 14). We want to highlight two key observations as follows:

**Firstly**, upon examination, we observe that despite the close proximity and smaller geographical distance between region 170 and region 164, they exhibit distinct urban functions. However, GraphST indicates their similarity. In contrast, the embedding vectors learned by our method, STLLM, successfully capture their differences, highlighting the effectiveness of our approach in capturing geographical semantics from a global perspective. **Secondly**, for the faraway region pairs, such as region 144 and region 14, the figure indicates that they share similar urban functions, which is reflected in the embedding vectors obtained from our method, STLLM. Conversely, GraphST fails to identify their similar functions. **In summary**, these observations validate the effectiveness of our STLLM in capturing global-view geographical semantics. This capability is likely achieved due to the successful expressive ability of LLM in capturing the global view.

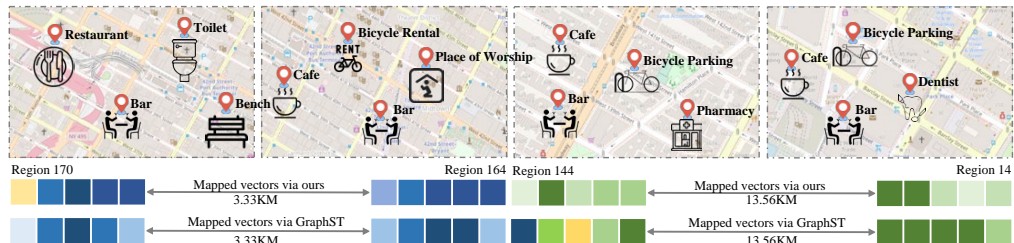

Figure 5: Case study of our STLLM method on New York City datasets.

## 5 CONCLUSION

This study highlights the potential of LLMs in enhancing spatio-temporal prediction and provides a comprehensive framework, STLLM, that integrates LLM-based knowledge learning with cross-view alignment for improved spatio-temporal understanding and forecasting. This simple yet effective paradigm captures spatio-temporal connections by aligning LLM-based knowledge representations with GNN-based structural embeddings, while also providing data augmentation and denoising capabilities. By incorporating urban semantics from a global view of LLM-augmented spatio-temporal knowledge, the framework successfully preserves both short-term and long-range cross-time and location dependencies in the latent representation space. The evaluation results of extensive experiments and comparisons with state-of-the-art baselines validate the effectiveness of the STLLM framework in achieving superior predictive performance. While our STLLM has exhibited impressive capabilities, their internal mechanisms remain opaque. Our future work involves in understanding and explaining the predictions of our LLM-based spatio-temporal learning in the context of natural language, so as to help practitioners make informed decisions and mitigate potential biases or errors.

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

## A  APPENDIX

### A.1  ALGORITHM OF STLLM

---

**Algorithm 1:** The STLLM Learning Algorithm

---

**Input:**  The spatial-temporal graph $\mathcal{G}$, the maximum epoch number $E$, the learning rate $\eta$;
**Output:** Regional Embedding $\mathbf{H}$
and trained parameters in $\boldsymbol{\Theta}$;

1  Initialize all parameters in $\boldsymbol{\Theta}$;
2  Design the spatial-temporal prompt $\mathcal{P}$;
3  Obtain the Embedding matrix $\mathbf{F}$ via LLM and $\mathcal{P}$;
4  Train the framework STLLM by Equation 6
5  **for** $epoch = 1, 2, ..., E$ **do**
6  $\quad$ Generate the subgraph $\mathcal{G}_1$ via teh random walk algorithm;
7  $\quad$ Send $\mathcal{G}_1$ and the corresponding adjacent matrix to GCN Encoder;
8  $\quad$ Obtain the embedding matrix $\mathbf{H}$ of N samples of subgraphs;
9  $\quad$ Minimize the loss $\mathcal{L}$ by Equation 6 using gradient decent with learning rate $\eta$;
10 $\quad$ **for** $\theta \in \boldsymbol{\Theta}$ **do**
11 $\quad\quad$ $\theta = \theta - \eta \cdot \frac{\partial \mathcal{L}}{\partial \theta}$
12 $\quad$ **end**
13 **end**
14 **Return $\mathbf{H}$** and all parameters $\boldsymbol{\Theta}$;

---

In this algorithm, the framework begins by designing a spatial-temporal prompt, which serves as input. The prompt is then passed into a Large Language Model (LLM) to generate an embedding matrix $\mathbf{F}$. Subsequently, a Graph Convolutional Network (GCN) encoder is trained to generate another embedding matrix $\mathbf{H}$ using the GCN equation 2 and optimizing it with the specified loss function 6. Steps 2 and 3 are repeated until convergence, ensuring that the resulting region embedding matrix is representative and captures the desired spatial-temporal information.

### A.2  DETAILED ANALYSIS FOR STLLM

In this section, we provide a comprehensive theory analysis based on the work of  Oord et al. (2018) on representation learning. This analysis forms the foundation for our approach of spatial-temporal graph contrastive learning using embeddings from a Large Language Model (LLM). The key concept is to leverage the principles outlined in  Oord et al. (2018)'s work to enhance the effectiveness of our spatial-temporal graph contrastive learning framework.

$$
\begin{aligned}
\mathbb{E}_{\tilde{\mathbf{H}}}[\log \frac{g(\mathbf{h}, \mathbf{f})}{\sum_{\mathbf{h}_i \in \tilde{\mathbf{H}}} g(\mathbf{h}_i, \mathbf{f})}] &\overset{g(\mathbf{h},\mathbf{f})=e^{G(\mathbf{h},\mathbf{f})}}{=} \mathbb{E}_{(\mathbf{h},\mathbf{f})}[G(\mathbf{h}, \mathbf{f})] - \mathbb{E}_{(\mathbf{h},\mathbf{f})}[\log \sum_{\mathbf{h}_i \in \tilde{\mathbf{H}}} e^{G(\mathbf{h}_i, \mathbf{f})}] \\
&= \mathbb{E}_{(\mathbf{h},\mathbf{f})}[G(\mathbf{h}, \mathbf{f})] - \mathbb{E}_{(\mathbf{h},\mathbf{f})}[\log(e^{G(\mathbf{h},\mathbf{f})}) + \sum_{\mathbf{h}_i \in \tilde{\mathbf{H}}_{\text{neg}}} e^{G(\mathbf{h}_i, \mathbf{f})}] \\
&\leq \mathbb{E}_{(\mathbf{h},\mathbf{f})}[G(\mathbf{h}, \mathbf{f})] - \mathbb{E}_{(\mathbf{h},\mathbf{f})}[\log \sum_{\mathbf{h}_i \in \tilde{\mathbf{H}}_{\text{neg}}} e^{G(\mathbf{h}_i, \mathbf{f})}] \\
&= \mathbb{E}_{(\mathbf{h},\mathbf{f})}[G(\mathbf{h}, \mathbf{f})] - \mathbb{E}_{\mathbf{h}}[\log \frac{1}{N-1} \sum_{\mathbf{h}_i \in \tilde{\mathbf{H}}_{\text{neg}}} e^{G(\mathbf{h}_i, \mathbf{f})} + \log(N-1)]
\end{aligned}
\tag{7}
$$

### A.3  DESCRIPTION OF BASELINES

We compare our model, STLLM, with baseline techniques from three research areas: graph representation, graph contrastive learning, and spatial-temporal region representation. This comprehensive analysis allows us to evaluate the strengths and advancements of our model in terms of graph representation, contrastive learning, and spatial-temporal information encoding.

Table 3: Data Description of Experimented Datasets

| Data | Census | Taxi Trips | Crime Data | POI Data | House Price |
|---|---|---|---|---|---|
| Description of Chicago data | 234 regions | 54,420 | 319,733 | 3,680,125 POIs (130 categories) | 44,447 |
| Description of NYC data | 180 regions | 128,566 | 60,002 | 20,659 POIs (50 categories) | 22,540 |

**Network Embedding/GNN Approaches**. We contrast our model STLLM with a number of typical network embedding and graph neural network models in order to assess its performance. To create region embeddings, we apply these models to our region graph $\mathcal{G}$. Following is a description of each baseline's specifics: **Node2vec** Grover & Leskovec (2016): Using a Skip-gram algorithm based on random walks, it encodes network structure information. **GCN** Kipf & Welling (2017): It carries out the convolution-based message transmission along the edges between neighbor nodes for embedding refinement. It is a graph neural design that permits information aggregation from the sampled sub-graph structures, as stated in the text after the GraphSage Hamilton et al. (2017). Graph Auto-encoder encodes nodes into a latent embedding space with the input reconstruction aim across the graph structures, according to **GAE** Kipf & Welling (2016). By distinguishing the degrees of significance among nearby nodes, the Graph Attention Network improves the classification capabilities of GNNs. **GAT** Veličković et al. (2018): By varying the relevance levels among nearby nodes, the Graph Attention Network improves the capacity of GNNs to discriminate.

**Graph Contrastive Learning Methods.** We compare our model STLLM with two graph contrastive learning models, and in addition to the aforementioned graph representation and GNN-based models, namely, **GraphCL** You et al. (2020): Based on the maximizing of mutual knowledge, this strategy generates many contrastive viewpoints for augmentation. The goal is to ensure embedding consistency across various connected views. **RGCL** Li et al. (2022): This cutting-edge graph contrastive learning method augments the data based on the intended rationale generator.

**Spatial-Temporal Region Representation Models.** We contrast it with contemporary spatial-temporal representation techniques for region embedding as well. The following are these techniques: **HDGE** Wang & Li (2017): It creates a crowd flow graph using human trajectory data and embeds areas into latent vectors to maintain graph structural information. **ZE-Mob** Yao et al. (2018): This method uses region correlations to create embeddings while taking into account human movement and taxi moving traces. **MV-PN** Fu et al. (2019): To represent intra-regional and inter-regional correlations, an encoder-decoder network is used. **CGAL** Zhang et al. (2019): An adversarial learning technique that takes into account pairwise graph-structured relations to embed regions in latent space. **MVURE** Zhang et al. (2021): In order to simulate region correlations with inherent region properties and data on human mobility, it makes use of the graph attention mechanism. **MGFN** Wu et al. (2022): In order to aggregate information for both intra-pattern and inter-pattern patterns, it encodes region embeddings with multi-level cross-attention. **GraphST** Zhang et al. (2023b): A robust spatial-temporal graph augmentation is achieved using this adversarial contrastive learning paradigm, which automates the distillation of essential multi-view self-supervised data. By enabling GraphST to adaptively identify challenging samples for improved self-supervision, it improves the representation's resilience and discrimination capacity.

### A.4 CATEGORY-SPECIFIC CRIME PREDICTION RESULTS

In the supplemental materials, we present the comprehensive evaluation findings on various criminal offense types for the cities of Chicago and New York in terms of the MAE and MAPE of 14 techniques. The foundational method for the other 14 methods is ST-SHN. The results in Table 4 demonstrate that our method STLLM consistently produces the best results on all crime categories for the two cities. This clearly demonstrates the substantial advantages that our region's embedding learning framework model brings. We credit the effectiveness of the spatial-temporal region graph's graph encoding in extracting useful regional features for region representation, as well as the various contrastive learning tasks, such as the contrastive learning paradigm for pulling close from the embedding matrix from LLM to that of GCN encoder. Besides, capturing global-view spatial-temporal graph knowledge via LLM is also beneficial to boosting the representation ability of our method.

### A.5 SPATIO-TEMPORAL PROMPT EXAMPLE

In this section, we provide an illustrative example of a spatio-temporal prompt, as depicted in Figure 7. This example highlights the effectiveness of incorporating spatial information in improving the

Table 4: Overall performance comparison in crime prediction on both Chicago and NYC datasets.

| Model | Chicago | | | | | | | | New York City | | | | | | | |
|---|---|---|---|---|---|---|---|---|---|---|---|---|---|---|---|---|
| | Theft | | Battery | | Assault | | Damage | | Burglary | | Larceny | | Robbery | | Assault | |
| | MAE | MPAE | MAE | MPAE | MAE | MPAE | MAE | MPAE | MAE | MPAE | MAE | MPAE | MAE | MPAE | MAE | MPAE |
| Node2vec | 1.1472 | 0.9871 | 1.7701 | 0.8945 | 1.9781 | 0.9764 | 1.9103 | 0.9712 | 4.8328 | 0.8572 | 0.6697 | 0.3974 | 1.1272 | 0.9566 | 0.9753 | 0.7020 |
| GCN | 1.1143 | 0.9675 | 1.3057 | 0.8123 | 1.5578 | 0.8126 | 1.5144 | 0.8173 | 4.7211 | 0.8428 | 0.6288 | 0.3470 | 1.0213 | 0.8626 | 0.7564 | 0.6637 |
| GAT | 1.1204 | 0.9708 | 1.3214 | 0.8408 | 1.5942 | 0.8231 | 1.5317 | 0.8188 | 4.7801 | 0.8215 | 0.6301 | 0.3518 | 0.9301 | 0.9293 | 0.7549 | 0.6329 |
| GraphSage | 1.1241 | 0.9765 | 1.3653 | 0.8609 | 1.6133 | 0.8643 | 1.5801 | 0.8506 | 4.7930 | 0.8448 | 0.6587 | 0.3952 | 0.9673 | 0.9056 | 0.7346 | 0.6423 |
| GAE | 1.1134 | 0.9675 | 1.3188 | 0.8193 | 1.5413 | 0.7998 | 1.4997 | 0.8066 | 4.7875 | 0.8395 | 0.6226 | 0.3504 | 0.9492 | 0.8643 | 0.7502 | 0.6308 |
| GraphCL | 1.0893 | 0.9012 | 1.0628 | 0.8419 | 1.3021 | 0.5261 | 1.2783 | 0.6429 | 4.3819 | 0.6528 | 0.6328 | 0.3562 | 0.7018 | 0.4312 | 0.6189 | 0.5503 |
| RGCL | 1.0790 | 0.8990 | 1.0567 | 0.8312 | 1.2078 | 0.5672 | 1.2084 | 0.6214 | 4.3792 | 0.6458 | 0.6450 | 0.3561 | 0.6901 | 0.4284 | 0.6184 | 0.5497 |
| HDGE | 1.0965 | 0.9123 | 1.0976 | 0.8005 | 1.3987 | 0.7304 | 1.3780 | 0.7367 | 4.5311 | 0.7582 | 0.6655 | 0.3916 | 0.8061 | 0.7049 | 0.7564 | 0.6637 |
| ZE-Mob | 1.1022 | 0.9604 | 1.3246 | 0.8309 | 1.5367 | 0.8201 | 1.5176 | 0.8284 | 4.5414 | 0.7523 | 0.6542 | 0.3870 | 0.7314 | 0.6944 | 0.7355 | 0.6401 |
| MV-PN | 1.0878 | 0.9201 | 1.1082 | 0.7906 | 1.4032 | 0.7405 | 1.3606 | 0.7245 | 4.4832 | 0.7360 | 0.6518 | 0.3831 | 0.7028 | 0.6871 | 0.7362 | 0.6399 |
| CGAL | 1.0896 | 0.9112 | 1.0876 | 0.7912 | 1.3986 | 0.7351 | 1.3607 | 0.7233 | 4.4935 | 0.7446 | 0.6564 | 0.3898 | 0.6958 | 0.5078 | 0.6572 | 0.6034 |
| MVURE | 1.0863 | 0.8932 | 1.0578 | 0.7983 | 1.3655 | 0.6382 | 1.2985 | 0.6607 | 4.4068 | 0.6663 | 0.6390 | 0.3708 | 0.6813 | 0.4677 | 0.6324 | 0.5882 |
| MGFN | 1.0824 | 0.8953 | 1.0765 | 0.7904 | 1.2943 | 0.5986 | 1.2507 | 0.6299 | 4.3767 | 0.6494 | 0.6595 | 0.3689 | 0.6901 | 0.4530 | 0.6278 | 0.5586 |
| GraphST | 1.0722 | 0.4764 | 0.8933 | 0.8424 | 1.1796 | 0.4387 | 0.9044 | 0.4714 | 4.3564 | 0.6455 | 0.6362 | 0.3430 | 0.6802 | 0.4035 | 0.6083 | 0.5337 |
| STLLM | **1.0717** | **0.4697** | **0.8392** | **0.7892** | **1.1651** | **0.4217** | **0.8940** | **0.4508** | **4.3430** | **0.6402** | **0.6213** | **0.3261** | **0.6766** | **0.3848** | **0.6028** | **0.5075** |

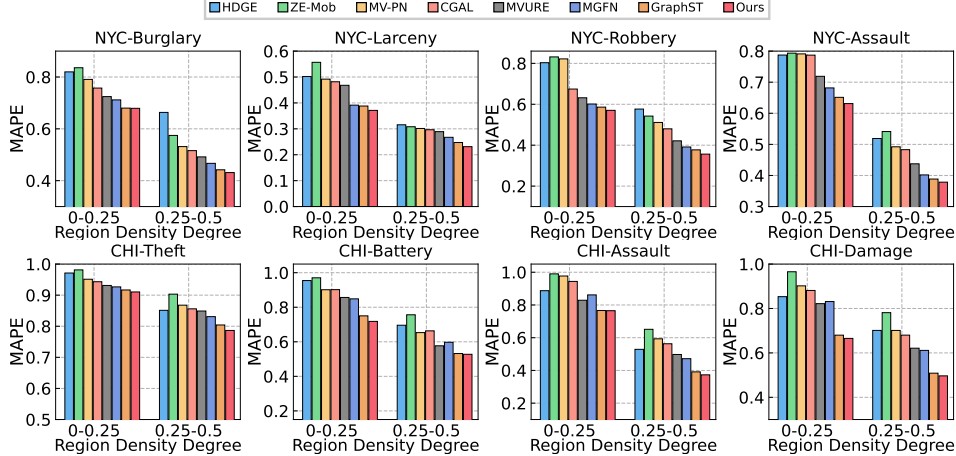

Figure 6: Results on NYC and CHI crime for four crime types *w.r.t* different data density degrees.

performance of summary generation. Additionally, we demonstrate how ChatGPT can generate summaries specifically tailored to the functions of different regions. **In the first case**, we showcase the impact of spatial information on summary generation. By considering the spatial context of the regions, we observe improved summarization results. Furthermore, ChatGPT leverages this spatial information to generate more informative summaries that capture the unique characteristics and functionalities of each region. **In the second case**, we investigate the influence of temporal information on the summarization process. By incorporating temporal dynamics, such as traffic flows and temporal semantics, ChatGPT is able to deduce additional connections and relationships among regions. This leads to more comprehensive and nuanced summaries that reflect the temporal aspects of the regions. **In summary**, by examining these cases, we highlight the significance of both spatial and temporal information in spatio-temporal prompt generation. The integration of these factors enhances the summarization capabilities of ChatGPT, allowing for more contextually-aware and informative summaries tailored to specific regions and their temporal dynamics.

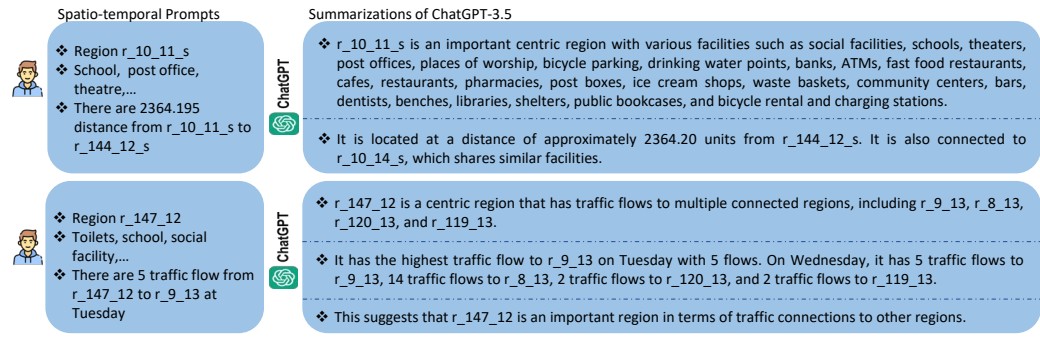

Figure 7: The spatio-temporal prompt example of STLLM.

