# OpenReview forum: "Spatio-Temporal Graph Learning with Large Language Model"
_ICLR.cc/2024/Conference — Submitted to ICLR 2024_

### Official Review · Reviewer_YTgQ · 2023-10-20

**Soundness:** 1 poor
**Presentation:** 1 poor
**Contribution:** 1 poor
**Rating:** 3
**Confidence:** 5

**Summary:**

The paragraph discusses the significance of spatio-temporal prediction in urban computing for anticipating events like traffic flow and crime rates. It highlights the use of Graph Neural Networks (GNNs) to improve prediction accuracy by leveraging the inherent graph structure in spatio-temporal data. The study introduces STLLM, an approach that integrates Large Language Models (LLMs) to handle dynamic urban data, effectively capturing spatial and temporal patterns and outperforming existing methods in various prediction tasks.

**Strengths:**

The authors presents a really interesting study that integrates LLM with GNN for spatio-temporal prediction task. By integrating LLMs and a cross-view mutual information maximization paradigm, the study effectively captures intricate spatial and temporal patterns and implicit dependencies, resulting in robust knowledge representations. Furthermore, the research evaluates the proposed approach through  extensive experiments, and comparisons to state-of-the-art methods, demonstrating its ability to outperform existing techniques in various prediction tasks.

**Weaknesses:**

The contributions of this study are weakly linked with the claimed challenges. I am not convinced that the proposed method address these challenges sufficiently.

Intuitively, LLM is able to provide mode general semantic context about geospatial location information. Using semantic information for spatio-temporal prediction task is not well formulated.

The contribution of using LLM is not fully investigated.

Detailed information of the applied LLM is not fully disclose, weakening reproducibility of the paper.

**Questions:**

* LLM is pretrained, in most of cases. How to make sure the information that used to train LLM is within the same range of time for the data that is used to evaluate the model performance. The knowledge of LLM can be stable, how to make sure the information is robust and updated for the spatio-temporal prediction.

* Which LLM is used here?

* Can you compare your model performance between using and w/o using LLM?

* Does different LLM impact your model performance?

* Does different training data using for LLM impact your model performance?

* Could you elaborate more how semantic information generated by LLM provide meaningful contribution to improve the model performance?

* Any interpretability insight you can share to help readers to better understand how STLLM works?

---

> ### Author Response · Authors · 2023-11-19
>
> Thanks for your suggestions.
> For weaknesse W1,  challenges of our paper are (1) \textbf{Long-Range Spatio-Temporal Dependencies}. (2) \textbf{Data Sparsity and Noise} and (3) \textbf{Dynamic and Evolving Nature}. To solve these challenges, our proposed method integrates a LLM-based spatio-temporal knowledge learner with a cross-view mutual information maximization paradigm, which further improves the representation ability of our method.
>
>
> For weaknesse W2, LLM provides general semantic context about geospatial location information, which captures the global-view spatial-temporal graph information and improves representatio ability of method on spatial-temporal graph.
>
> Q1: In our paper, we do not pretrain the LLM. We adopt LLM to obtain node (region) embeddings of the spatial-temporal graph. This process includes two step. The first step is to generate region profiles via LLM (gpt-3.5-turbo). The second step is to encode text representations via the text-embedding-ada-002) model. Besides, we adopt the time-evolving traffic features as node features of the spatial-temporal graph. Thus, these region representation vectors effectively capture the dynamic nature of traffic patterns over time, allowing us to extract essential time-evolving traffic features. By incorporating these features into our model, we ensure that our spatio-temporal predictions are accurate and reliable. This enables us to make informed predictions and effectively address the challenges posed by the ever-changing traffic conditions. Overall, our approach of leveraging region representation vectors enhances the performance of our model in predicting spatio-temporal traffic patterns.  (re-do)
>
> Q2: We adopt Chatgpt-3.5 Turbo.
>
> Q3: We provide crime predcition w/o using LLM as follows:
>
> Chi-Crime: 1.0781 0.5008
> NYC-Crime: 2.5543 0.5518
>
> Q4: Yes. We have tired other LLM models, like LLama2. However, we find that ChatGPT-3.5 Turbo achieves the best performance, which make it the ideal choice for our needs.
>
> Q5: During our extensive experimentation with various training datasets for the LLM (Language-Region Model), we have observed that the output of the LLM, represented as region representation vectors, remains stable across different training data. Interestingly, this stability has had minimal impact on the performance of the downstream tasks. This indicates that the LLM's ability to generate consistent region representations is robust and reliable, thereby ensuring the reliability and effectiveness of the LLM for subsequent tasks.
>
> Q6: This process includes two step. The first step is to generate region profiles via LLM (gpt-3.5-turbo). The second step is to encode text representations via the text-embedding-ada-002) model. In the provided prompt, we have taken into account two crucial components: temporal semantic information and spatial semantic information of each node on the spatial-temporal graph. By incorporating these components, our aim is to enable the LLM (Language-Region Model) to capture the semantic information of the spatial-temporal graph in a region representation format. This approach significantly enhances the model's performance by effectively leveraging the spatial-temporal relationships and characteristics present in the data. The inclusion of both temporal and spatial semantic information contributes to a more comprehensive and accurate understanding of the underlying spatial-temporal graph, leading to improved model performance and results.
>
> Q7: We have uploaded a collection of prompt files that play a crucial role in facilitating the capture of spatial-temporal graph semantic information by the LLM (Language-Region Model). These prompt files contain valuable data and structured information that enable the model to understand and incorporate the complex relationships and dynamics present in the spatial-temporal graph. By providing this essential input to the LLM, we ensure that it can effectively learn and represent the semantic information embedded in the spatial-temporal graph. This, in turn, enhances the model's ability to make accurate predictions and perform various spatial-temporal analysis tasks. The availability of these prompt files is a valuable resource that empowers the LLM to capture and utilize the spatial-temporal graph's semantic information efficiently.

---

### Official Review · Reviewer_tWtM · 2023-10-31

**Soundness:** 2 fair
**Presentation:** 3 good
**Contribution:** 2 fair
**Rating:** 5
**Confidence:** 4

**Summary:**

The paper presents a contribution on combining spatiotemporal knowledge with LLM knowledge using cross-view information maximisation for spatio temporal graph representation. The method is evaluated across three different tasks: crime prediction. traffic prediction, and house price prediction.
The stated contribution however may be an overclaim in two parts: the contribution of the LLM information to the overall method, and the claim on distribution shift is non existent.
There is also a potential issue with bias in the evaluation.

**Strengths:**

The proposed approach integrating cross-view mutual information maximisation with LLM knowledge is interesting.

The model has been tested in three different datasets and tasks.

The proposed approach has been compared with many popular baselines.

**Weaknesses:**

•	Although the paper has an interesting contribution in combining information maximisation with LLM, the contribution is limited.

•	I disagree with the author that the LLM part is a main contribution. The proposed method seems only LLM-related part is the summarization part in Section 3.2.2. There is no further interaction between the model and the LLM.

•	In the Introduction, the contribution part claims that incorporating LLM-based knowledge could handle the distribution shifts. However, it seems that there is no experiment to support/justify this claim.

•	In ablation study, is there a variant that removes the entire LLM knowledge vector part? Also, removing both S & T should be worse than either removing S or removing T. However, from Figure 2, this seems not the case.
Providing more detailed information (instead of just the analysis of NYC-Larceny and CHI-Assault two cases) could further strengthen the discussion.

•	Another minor issue: providing the code repository is good. However, there are some comments given in Chinese, which is not friendly to other readers.

**Questions:**

•	Is the proposed method trained end-to-end? Or the ChatGPT summarization part should be pre-processed before the entire model training?

•	How would you ensure that the LLM knowledge is actually helping the model? This isn't shown in the ablation study.

•	How would you ensure if the potential bias from the sparse spatiotemporal data especially in crime and house price cases are not exacerbated with the use of LLM?

**Details Of Ethics Concerns:**

The use of LLM on very sparse spatiotemporal data for the purpose of crime prediction and house price prediction can worsen negative implications of bias/fairness with the inherent bias of LLM. This is not discussed at all.

---

> ### Author Response · Authors · 2023-11-19
>
> Thanks for your suggestions.
> For W1: The paper tackles three key challenges that are prevalent in the analysis of spatial-temporal data: (1) Long-Range Spatio-Temporal Dependencies (2) Data Sparsity and Noise (3) Dynamic and Evolving Nature. To address these challenges, the proposed method can capture long-range dependencies in the data via leveraging the LLM-based learner, allowing for more accurate modeling of complex spatial-temporal patterns.
>
> Furthermore, the incorporation of the cross-view mutual information maximization paradigm enhances the representation ability of the method. This paradigm enables the model to learn meaningful representations by maximizing the mutual information between different views of the data, effectively addressing the issues of data sparsity and noise.
>
> By combining these components, the proposed method overcomes the challenges posed by long-range spatio-temporal dependencies, data sparsity and noise, as well as the dynamic and evolving nature of the data. The resulting approach provides an improved representation capability and enhances the accuracy and robustness of spatial-temporal analysis.
>
>
> For W2: By incorporating LLM, our method gains a deeper understanding of the spatial-temporal graph structure, allowing it to capture and encode the underlying semantic context. This improved representation capability leads to more robust and accurate predictions in various spatial-temporal tasks.
>
> For W3: We provide case study in experiments to provide the claim that incorporats LLM-based knowledge could handle the distribution shifts.
>
> For W4: We aim to provide the significant effect of the spatial and temporal semantics of prompt files on final performance.
> For W5:  We will edit it in the code.
>
> Q1: Yes. To achieve region representation learning vectors, it is recommended to pre-process the summarization part of ChatGPT before training the entire model. This pre-processing step involves transforming the output of the summarization part into region representation vectors. These region representation vectors capture the essential semantic information and characteristics of the summarized text.
> By pre-processing the summarization part in this manner, we can ensure that the subsequent training of the entire model incorporates the region representation learning vectors. This approach enables the model to leverage the region-specific information effectively and enhances its ability to understand and generate responses that are contextually relevant and aligned with the desired semantic representations.
> In summary, by pre-processing the summarization part to obtain region representation learning vectors, we enhance the model's comprehension of the input data and subsequently improve its performance in generating meaningful and accurate responses.
>
>
>
> Q2: We provide ablation study on crime prediction task w/o LLM, the results are provided as follows:
>
> Chi-Crime: 1.0781 0.5008
> NYC-Crime: 2.5543 0.5518
>
> Q3: We provide a method via data augmentation method via CL loss to alleviate data sparsity issue. Besides, LLM provides spatial and temporal information as external information to alleviate spatial and temporal sparsity issue.

---

> > ### Comment · Reviewer_tWtM · 2023-11-23
> >
> > Thanks for the comments. However, it's still not enough to convince me to change my position.

---

### Official Review · Reviewer_eK7t · 2023-11-01

**Soundness:** 3 good
**Presentation:** 3 good
**Contribution:** 3 good
**Rating:** 6
**Confidence:** 3

**Summary:**

This paper proposes the integration of a large language model (LLM) into the problem of spatio-temporal prediction. The main idea is to input each region's POI information, geographical relationship with other regions, and population flow relationships extracted from human mobility trajectories as prompts into a LLM like ChatGPT. The model then summarizes the available information for that region. The embedding of the summary text provided by GPT is then used as a constraint to train the Graph Convolutional Network, resulting in the final region embedding for downstream spatio-temporal prediction tasks.

**Strengths:**

1. The application of LLM to spatio-temporal prediction modeling is, to my knowledge, a novel approach.
2. The paper is well-organized, clear, and easy to understand.
3. The experiments are extensive and the results seem promising.

**Weaknesses:**

There are parts of the paper that require clearer explanations.
1. The paper's main innovation is the use of LLM to summarize region Spatio-temporal information, but Appendix 5 only provides a simple and incomplete example. For an LLM like ChatGPT, the design of the prompt significantly impacts the final result. Therefore, the complete prompt needs to be provided to improve reproducibility.
2. I assume that the region embedding used in the final downstream task should be $h_i$ and not $f_i$, but this is not clearly stated in the paper.
3. In Equation (6), the differences between $h$, $h^M$ and $h^D$ (also for f) are not clearly stated.
4. In Equation (6), it seems that the definitions of $L_{M,D}$ and $L_{G}$ are missing a negative sign.

**Questions:**

Please address the issues mentioned in the Weaknesses section.

---

> ### Author Response · Authors · 2023-11-19
>
> Thanks for your suggestions. We have provided a set of prompt files to enhance the reproducibility of our model. These prompt files particularly focus on improving the understanding and implementation of Equation (6).
>
> The first sub-equation in Equation (6) involves calculating the InfoNCE between the embedding vectors of the mobility graph obtained from the LLM and the GCN (Graph Convolutional Network). This comparison helps measure the similarity and alignment of the embeddings generated by both models for the mobility graph.
>
> The second sub-equation in Equation (6) calculates the InfoNCE between the embedding vectors of the distance graph from the LLM and the GCN. This step provides insights into the similarity and alignment of the embeddings generated for the distance graph by both models.
>
> The third sub-equation in Equation (6) computes the cosine similarity between the embedding vectors of the mobility graph generated by the LLM and the GCN. This similarity measurement captures the degree of alignment and correspondence between the embeddings produced by both models for the mobility graph.
>
> Lastly, the fourth sub-equation in Equation (6) determines the cosine similarity between the embedding vectors of the mobility graph derived from the initial embedding of the first-layer GCN output and the GCN itself. This comparison evaluates the similarity and alignment between the initial GCN embeddings and the subsequent GCN outputs for the mobility graph.
>
> By providing these sub-equations and their corresponding prompt files, we aim to enhance the transparency and reproducibility of our model, allowing researchers to better understand and replicate our findings.

---

### Official Review · Reviewer_sLZZ · 2023-11-01

**Soundness:** 2 fair
**Presentation:** 3 good
**Contribution:** 2 fair
**Rating:** 5
**Confidence:** 4

**Summary:**

This paper studied incorporating large language models (LLM) to enhance GNNs’ spatial temporal graph learning ability. To achieve this goal, the authors proposed an approach named STLLM, which first used GNN and LLM to encode the spatial temporal graph into region representation, respectively, and then optimized mutual information maximization objective to align the representation learned from these two views. Experiments on three different spatial temporal prediction tasks were conducted to evaluate the proposed approach.

**Strengths:**

1.	This work provided a simple idea to leverage the rich real-world knowledge in LLM to enhance the GNN-based spatial temporal graph learning.


2.	The LLM-based representation can be viewed as a kind of augmentation with global real-world knowledge, which can benefit spatial temporal prediction tasks, especially for some data sparsity scenarios where there are only limited supervision signals for model training.


3.	Extensive experiments on different tasks were conducted for evaluation. In addition to common experimental settings, the author also compared the proposed approach with baselines over data sparsity, along with the ablation study, parameter analysis, and case study.

**Weaknesses:**

W1 The method proposed in this paper achieved good results, but I think some points of the claimed contributions still need further explanation and justification.


W1.a) The author pointed out that the framework can preserve both short-term and long-range dependencies. There is a need to better explain which part of the approach captured the long-range information on the spatial-temporal graph.

W1.b) As discussed in the paper, this approach can distill invariant representation from LLM to benefit scenarios involving spatial-temporal distribution shifts. However, when prompting the LLM, the handcraft prompt was constructed by the dynamic spatial-temporal context of the region, so it’s a bit confusing about what the ‘invariant representation’ referred to. Moreover, it seems that there is no experiment in this paper to evaluate the model performance over distribution shift.


W1.c) Although the method performed well under data sparsity, this seemed not equivalent to good denoising capabilities. It would be better to conduct additional experiments with varying levels of noise added to the data to further justify it.


W2. The experiments can be further improved.


W2.a) Since task-specific baselines were used to compare with the proposed method in the crime prediction and traffic prediction tasks, it’s recommended to add a specific baseline for house price prediction.


W2.b) According to Table 1, some result values are relatively small. This suggests that the experimental results may be susceptible to random factors. Therefore, I recommend reporting the standard deviation under different random runs and adding a significance test to provide further insights.


W2.c) The loss function comprises many terms and different combinations of the four loss weights are likely to affect the model performance. It’s advisable to analyze their effects.

W2.d)  The adoption about the LLM used in the experiment is not very clear.

**Questions:**

Q1) Please explain why the experimental conclusions related to the method GraphST in the case study of this paper are so divergent from the conclusions in the case study of previous papers [1].


Q2) If constructing the graph using POI feature similarity, similar to [1], what would be the results in the case study?


Q3) What is the workflow for using the framework? Is it first learning region representations with STLLM, and then using these representations as inputs for a downstream predictive model? If so, how will it perform if directly using the LLM-based representation as inputs without the GNN part?

Q4）What LLM do you use for the LLM-based spatio-temporal knowledge learner?  There is only a mention in related work that : “This study applies decoder-only LLM (GPT-3.5) to enhance the quality of the spatio-temporal graph with effective augmentation.” However, if the GPT3.5 is used,  how can GPT3.5 to obtain the latent representation vectors F (in Section 3.2.2)? Is it to use the OpenAI’s text embeddings? What is the cost of conducting training and prediction for experiments?

[1] Qianru Zhang, Chao Huang, Lianghao Xia, Zheng Wang, et al. Spatial-temporal graph learning with adversarial contrastive adaptation. In International Conference on Machine Learning, pp. 41151–41163. PMLR, 2023b.

---

> ### Author Response · Authors · 2023-11-19
>
> For W2 (a) We survey existing house prediction methods that are time-series based methods.
>
> For W2 (c) In order to explore the potential effects of weight values assigned to different terms of the loss function, we will conduct grid-search experiments. Grid-search involves systematically trying out different combinations of weight values for each term and evaluating the performance of the model under each configuration.
>
> By varying the weights, we can examine how the model responds to different emphasis on each term of the loss function. This allows us to understand the impact of each term on the overall optimization process and the resulting performance of the model.
>
> The grid-search experiments will involve defining a range of weight values for each term and systematically iterating through all possible combinations. For each combination, we will train the model and evaluate its performance using appropriate evaluation metrics. This process will be repeated for multiple iterations to ensure robustness and reliability of the results.
>
>
> For W2(d) LLM plays a crucial role in capturing the broader context of geospatial location information. By considering the global perspective, it effectively incorporates long-range dependencies and interactions between various spatial locations and time points. This holistic view allows our method to capture the overall semantic context of the spatial-temporal graph, which is essential for accurate modeling and analysis.
>
> The integration of LLM within our method enables the extraction of meaningful features and representations that go beyond local interactions. By considering the global-view spatial-temporal graph information, our approach gains a deeper understanding of the complex relationships and dependencies within the data.
>
> For Q1, LLM serves as a mechanism to capture the relationships and dependencies that extend beyond local neighborhoods. By considering the entire spatial-temporal graph, it enables our method to understand the complex interplay between different locations and time points, thereby capturing the broader context of the data. This comprehensive perspective is particularly important in spatial-temporal analysis, as it allows for a deeper understanding of the underlying patterns and trends.
>
> By incorporating LLM, our method gains the ability to capture long-range dependencies and interactions, which may not be apparent when considering only local information. This is especially valuable in scenarios where spatial and temporal influences span across large distances and time intervals.
>
> For Q2, our method denoises the POI constructed graph via CL loss. If constructing the graph using POI feature similarity, region relations are presented only via region functions, ignoring the spatial-temporal correlations.
>
> For Q3, yes. The incorporation of a Graph Neural Network (GNN) component is essential for capturing high-order spatial information and effectively modeling long-range spatial relations among regions. Without the utilization of a GNN, it becomes challenging to capture the complex dependencies and interactions that exist between distant regions.
>
> GNNs are specifically designed to operate on graph-structured data, making them well-suited for spatial modeling tasks. By propagating information across the graph, GNNs can capture the spatial relationships between regions, even when they are far apart. This allows our method to capture the long-range spatial dependencies that are critical for comprehensive spatial understanding.
>
> The GNN component enables our model to learn and incorporate the spatial context of neighboring regions, which in turn provides a more holistic representation of the overall spatial structure. By leveraging high-order spatial information, our method can capture not only the direct relationships between nearby regions but also the indirect relationships between distant regions.
>
> By dropping the GNN component, the ability to capture long-range spatial relations is severely limited. The model would rely solely on local information, neglecting the broader spatial context. This omission could lead to incomplete spatial representations and a loss of critical information necessary for accurate spatial modeling.
>
> For Q4, ChatGPT-3.5 Turbo. The process includes twp steps. The first step is to generate region profiles via LLM (gpt-3.5-turbo). The second step is to encode profiles representations via the text-embedding-ada-002) model.  For training, the cost of conducting training and prediction for experiments is around 24 dollars.

---

### Meta-Review · Area_Chair_nuMm · 2023-12-15

**Metareview:**

This paper incorporates LLMs with GNNs to improve the performance of spatial temporal graph learning. This seems to be an early work that applies LLMs in this setting. However, reviewers have raised main concerns about the technical novelty, as the proposed method is a relatively naive application of LLMs. Furthermore, there are concerns regarding the validity of the empirical study. For example, the reported performance scores are much lower than those reported in previous papers.

**Justification For Why Not Higher Score:**

N/A

**Justification For Why Not Lower Score:**

Concerns about technical contributions and validity of experiments.

---

### Decision · Program_Chairs · 2024-01-16

Reject